# Effects of improved on-farm crop storage on perceived stress and perceived coping in pregnant women—Evidence from a cluster-randomized controlled trial in Kenya

Heike Eichenauer[1], Matthias Huss[2,3], Michael Brander[2,3], Thomas Bernauer[2], Ulrike Ehlert[1] *

1 Clinical Psychology and Psychotherapy, University of Zurich, Zurich, Switzerland, 2 Institute of Science, Technology and Policy (ISTP), ETH Zurich, Zurich, Switzerland, 3 Zurich Knowledge Center for Sustainable Development & Informatics and Sustainability Research Group, University of Zurich, Zurich, Switzerland

* u.ehlert@psychologie.uzh.ch

## Abstract

### Background

Food insecurity can be harmful to pregnant women, as pregnancy is a challenging period with increased maternal nutritional requirements to ensure optimal fetal development and health of the mother. Whether food insecurity negatively affects maternal health may depend on how stressful pregnant women perceive this food insecurity to be and how strongly they believe they can cope with it. In Sub-Saharan Africa (SSA), pregnant women from smallholder households suffer from food insecurity due to post-harvest losses (PHL), i.e., loss of crops because of inadequate storage. An agricultural intervention that improves crop storage has been shown to reduce food insecurity. However, it remains to be determined whether this agricultural intervention (treatment) has an additional positive effect on pregnant women's perceived stress levels and coping abilities. This study examines whether pregnant women from treatment households experience lower perceived stress levels and higher perceived coping abilities compared to pregnant women from control housholds.

### Methods and findings

In a randomized controlled trial (RCT), short message service (SMS)-based mobile phone surveys were conducted to assess the causal effect of a food security intervention (improved on-farm storage of maize) on perceived stress and coping in pregnant women from smallholder households. Pregnant women were identified through these monthly surveys by asking whether someone in their household was currently pregnant. The significant results revealed that pregnant women from treatment households experienced more perceived stress but better perceived coping abilities compared to pregnant women from control households. Uncertainty due to lack of experience, this might have contributed to the higher perceived stress, as the women could not easily judge the benefits and risks of the

---

**Data Availability Statement:** Our application to the ethics committee (IRB) did not include an explicit request to make the study data publicly available.

However, anonymized data will be available upon request to the Ethics committee of ETH Zürich for study replication (Contact details are as follow: Sekretariat Ethikkommission, E-mail: ethics@sl. ethz.ch) or upon request to the corresponding author (Prof. rer. nat. Ulrike Ehlert, E-mail: u. ehlert@psychologie.uzh.ch).

**Funding:** This research was funded by Stiftung Dreiklang and Stiftung Fons Margarita and three anonymous foundations. The funders had no role in study design, data collection and analysis, decision to publish, or preparation of the manuscript.

**Competing interests:** The authors have declared that no competing interests exist.

new storage technology. However, the technology itself is a tangible resource which might have empowered the pregnant women to counteract the effects of PHL and thus food insecurity.

## Conclusion

Our findings indicate that pregnant women from treatment households had higher perceived coping abilities but experienced more perceived stress. More research is needed on how this technology impacts maternal mental health in a broader sense and whether biological mechanisms, such as epigenetics, may underlie this association.

## Introduction

Adverse environmental conditions during pregnancy, such as food insecurity, can negatively impact the psychological and physiological health of women as well as the development of the fetus [1–3]. Food deprivation can be particularly harmful to pregnant women in low-income countries, as pregnancy is a very challenging period with increased maternal nutritional requirements to ensure optimal fetal development and health of the mother [4–6]. Research has shown that undernutrition during pregnancy is associated with an increased risk of mortality and morbidity for both mother and offspring [4, 7]. Food insecurity can lead to adverse perinatal outcomes such as low birth weight and preterm birth [8] as well as pregnancy complications such as inadequate pregnancy weight gain, hypertension, preeclampsia, gestational diabetes [4, 9], and mental disorders in the mother [8].

Whether food insecurity during pregnancy provokes negative maternal health outcomes can depend on how stressful pregnant women perceive this environmental condition to be. According to Lazarus and Folkman's transactional stress theory [10, 11], an individual is in constant exchange with the environment through two cognitive appraisal processes, thus influencing his/her well-being. First, the individual determines whether a particular environmental circumstance is irrelevant, positive, or stressful to him or her. The evaluation of the environment as stressful depends on whether it is perceived as harmful, threatening, or challenging (primary appraisal). The primary appraisal of a stressful environment is followed by a secondary appraisal, in which the individual determines whether he or she has adequate resources and coping strategies to deal with the environment. Harmful and threatening environments are often accompanied by feelings of uncontrollability over the situation and may thus trigger negative emotions [10–12] (for more detailed information, see this review [13]). With regard to food insecurity, numerous studies have reported a positive association between increased food insecurity and impaired mental health in African populations [14]. For instance, pregnant Ethiopian women living in food-insecure households were four times more likely to experience psychological distress than their food-secure counterparts [15]. Psychological distress is an important component of overall mental health and a risk factor for depression [16–18]. Researchers have demonstrated that food-insecure pregnant women in Africa are up to five times more likely to suffer from depression [19, 20]. In the context of the country examined in this study, Kenya, there is evidence of a positive association between food insecurity and depressive symptoms in perinatal women [21, 22]. Specifically, the odds of suffering from depressive symptoms increased almost six fold when food insecurity was high [21]. In addition, perceived stress was higher among Kenyan postpartum women who experienced moderate food insecurity than among Kenyan postpartum women who experienced mild food insecurity [23].

In Sub-Saharan Africa (SSA), food insecurity is particularly high among smallholder farming households, attributable to factors such as climate change-related extreme events and seasonality, which hamper agricultural production [24–27]. For example, a 10-day delay in the onset of the rainy season increases the risk of seasonal drought, thus affecting the food security status of smallholder farmers in SSA [28]. Furthermore, heavy rainfalls can prevent smallholder farmers from adequately drying their crops before storage, which can lead to increased moisture and thus mold growth during the storage period, rendering the crop inedible and therefore lost [29–31]. Extreme weather shocks, such as the aforementioned drought and rainfall, negatively affect crop production in Kenya [32] and are a major concern for smallholder farmers in rural Kenya [33]. Post-harvest losses (PHL) refer to the amount of crop lost due to factors such as crop handling and crop storage [31], with the latter accounting for the majority of PHL [34, 35]. Indeed, a meta-analysis revealed that on average, 25.6% of harvested maize in SSA is lost due to insufficient crop storage [35]. A recent article states that the PHL of maize by storage pests amount to about 36% overall, with Western Kenya being the most affected [36]. Moreover, the lean season, i.e. a few months after the harvest, when food stocks are depleted and the coming crops were yet to be harvested, poses the challenge of maintaining food security in smallholder farming households [27, 37], which can be further exacerbated by PHL.

Agricultural interventions that focus on PHL, a well-known contributor to food insecurity, may be helpful to counteract the negative effects of food insecurity on maternal mental health and the intergenerational adverse health effects for the offspring. One method to improve crop storage is the distribution of hermetic storage bags instead of conventional polypropylene bags. Hermetic bags have been shown to improve crop storage in Tanzania and thus reduce PHL significantly [38]. A reduction in food insecurity due to hermetic storage bags was also demonstrated among Kenyan smallholder farmers from the project described in the present study [39]. However, its effect on perceived stress and perceived coping (two components of the transactional stress theory) in pregnant women still needs to be elucidated.

In summary, on the correlation level, numerous studies have shown that food insecurity is associated with poorer mental health outcomes. We take this research forward by determine whether an agricultural intervention to improve food security (allocation and use of hermetic storage bags in households) can have beneficial mental health outcomes for pregnant women from smallholder households. We hypothesized that pregnant women in the treatment households would experience lower perceived stress and higher perceived coping as compared to pregnant women in the control households. Following the randomly allocated intervention, we collected perceived stress and perceived coping data over several months using monthly short message service (SMS)-based surveys. The treatment effects were analyzed via Intention-to-treat (ITT) analyses.

## Methods

### Study setting

The present randomized control trial (RCT) was conducted in Kakamega County, Western Kenya and included all 12 sub-counties of Kakamega and 59 out of 60 wards. Smallholder farming, and maize as the main staple food, are dominant in this Sub-Saharan region. The study area is affected by seasonality, with the lean season typically beginning around April or May and a secondary smaller harvest taking place around January or February [39]. A map of the study region and the smallholder farming groups' locations can be found in Huss et al. [39].

## Research design and intervention

This study was embedded in a matched-pair, cluster-randomization design. Pair-wise matching of the farmers was performed using baseline variables such as food security and mean market distance [40]. After pairing, farmers were randomly assigned to the treatment or control group using spatial clustering. This resulted in 62 experimental clusters with 285 farmer groups (5'444 smallholder households). More detailed study information can be found in Huss et al. [39]. Of these 5'444 smallholder households, a subsample of 1,591 pregnant women took part in the present study.

The treatment group received five hermetic storage bags of the brand "AgroZ" per household and standardized training in their usage. Each bag has the capacity to store 100kg of maize. The intervention was implemented from 3[rd] to 15[th] September 2019 by our local partner International Centre of Insect Physiology and Ecology (*icipe*), with the training session developed by some of the authors of this paper. The control group continued storing their maize as usual–mostly in traditionally used polypropylene bags.

## Outcome variable

The RCT was designed to assess the effects of the intervention (hermetic storage bags) on food security and a variety of health outcomes [41]. The present study focused on perceived stress and perceived coping, measured by the short form of the Perceived Stress Scale (PSS-4). The PSS is based on the stress theory of Lazarus [10, 12] and measures the degree to which an individual perceives his/her life as stressful as well as the individual's perceived ability to cope with stress. The original version of the PSS consists of 14 items [42]; nowadays, a 10-item version is commonly used, and the scale has been further reduced to four items [43]. While some researchers found that the PSS-14, -10 and -4 capture two distinct psychological factors, *coping* and *stress* [44–49], others challenge this notion [50, 51], particularly for the PSS-4 [52, 53]. The PSS questionnaires have acceptable psychometric properties and have been validated in different countries (see review, [46]), as well as for pregnant women [54–56]. As the PSS-10 has been widely used to study Kenyan populations [23, 57–62], it is reasonable to assume that the PSS-4 can also be applied in Kenya. Our Kenyan partner *icipe* translated and back-translated the PSS-4 into Kiswahili and English (see S1 Table for the final translation of the PSS-4). We analyzed the PSS-4 data as two distinct psychological factors: The psychological factor "perceived coping" consists of items 2 and 3 of the PSS-4 and measures the perceived ability to cope with stress. The other factor, consisting of items 1 and 4, asks about the amount of stress the respondent is experiencing and is referred to as "perceived stress". Higher sum scores for the perceived stress factor indicate higher stress and higher sum scores for the perceived coping factor indicate a lower perceived ability to cope with stress. With regard to internal consistency in our data set, Cronbach's alpha for the one-factor model was less than 0.6 at 11 out of 12 measurement time points. For the two-factor model, the perceived stress factor had a Cronbach's alpha greater than 0.6 at eight out of the 12 time points and the perceived coping factor had a Cronbach's alpha greater than 0.6 at nine out of the 12 time points. Additionally, we analyzed the psychometric properties of the PSS-4 using a principal component analysis (PCA) for each measurement time point. The one-factor model explained a maximum of 47% of the variance while the two-factor model explained over 66% of the variance for each measurement time point (S2 and S3 Tables), thus supporting our choice of the two-factor model.

## Data collection method

Throughout the RCT, the smallholder farmers received monthly SMS-based mobile phone surveys asking about their food security status [39]. Within these monthly surveys, the farmers

were also asked whether someone in their household was currently pregnant. Farmers who answered "yes" to this question received an additional SMS-based mobile phone survey containing the PSS-4 questionnaire. Pregnant women could fill out the PSS-4 questionnaire via SMS-based survey every month while they were still pregnant. As soon as farmers answered "no" to the pregnancy question, they were excluded from the subsequent PSS-4 survey rounds. The observation period for this analysis was monthly from April 2020 until February 2021. The PSS-4 was seen as most suitable for use in SMS-based surveys as longer versions of the PSS are inefficient for repeated SMS-based surveys in larger populations in a developing country context [42]. For each completed survey round, participants received telephone credit (airtime) valued at 10 Kenyan Shilling. The treatment and control group received the same amount of airtime.

## Research aim and statistical analysis

The purpose of this study was to determine whether an agricultural intervention consisting of an improved on-farm storage technology with hermetic storage bags affects the perceived stress levels and coping abilities of pregnant women in Kenyan smallholder households. As an alternative hypothesis, we also explore whether pregnant women in the treatment households have lower perceived stress and higher perceived coping abilities compared with pregnant women in the control households. Our null hypothesis is that there are no differences between perceived stress and coping by pregnant women in treatment and control households.

ITT analyses was used to analyze treatment effects (allocation and use of hermetic storage bags in the household) on perceived stress and perceived coping in pregnant women from smallholder farming households. ITT analysis is a recommended approach in RCTs when compliance and adherence to interventions cannot be guaranteed [63, 64]. The ITT was estimated as the weighted average of the mean differences within pairs between treatment and control clusters [65] of pregnant women. For this purpose, arithmetic weights (wk) = n1k + n2k was used, representing the sum of the n observations in both clusters of each pair indexed by k. For further information see Imai and colleagues [65]. In addition, linear mixed-effects models was estimated for the entire observation period for both outcome measures, perceived stress and perceived coping, with the dichotomous variable treatment (yes/no) as the fixed effect and a random intercept that varied across the paired IDs (PID). Analyses were conducted using MATLAB and the open-source software R version 4.1.0 (the R Foundation for Statistical Computing, Vienna, Austria).

## Ethics approval

Ethics approval was granted by the ETH Zurich Ethics Commission (EK, 2018-N-51) and *icipe's* Science Committee (no approval number used). Registration of the study design can be found in the American Economic Association (AEA) RCT Registry [41]. Informed written consent of the households to participate in the study was obtained before the start of the intervention.

## Results

### Sample characteristics

During the observation period of April 2020 to February 2021, a total of 1,591 pregnant women from smallholder households participated in the SMS-based survey. Sample characteristics of baseline variables can be found in S4 Table. Baseline variables did not differ between control and treatment households, with the exception of slight differences in household size

and age of household head. The households of the control group were slightly larger (M = 6.66, SD = 2.52) compared to those of the treatment group (M = 6.4, SD = 2.35) and the household heads of the control group (M = 45.21, SD = 12.78) were slightly older than the household heads of the treatment group (M = 44.14, SD = 12.35). However, these differences in household size (0.26) and age of household head (1.07 years) were very small.

## Perceived stress and perceived coping in pregnant women from control households

In pregnant women in the control households, perceived coping scores were relatively stable during the lean season (between *M* = 4.4 and *M* = 4.6). After the end of the main harvest of 2020, perceived coping improved (lower mean values) for nearly all consecutive months (Fig 1). Perceived stress levels in pregnant women in the control households fluctuated over 12 consecutive months and were, overall, lower during the lean season. Compared to the lean season, pregnant women in the control households generally felt more stressed after the main harvest (Fig 2).

## Impact of improved on-farm storage on perceived coping in pregnant women

For perceived stress and perceived coping, we expected the strongest treatment effects during the lean season when food stocks were depleted, which would then decrease after the start of the new storage period (the end of the main harvest). The results of the linear mixed-effects model show that pregnant women in the treatment households scored on average 0.167 points less than the control households on the coping measure over the entire observation period combined, which indicates higher perceived coping (*t*(6082) = -3.179, *p* = 0.001).

When analyzing each survey month separately, pregnant women in the treatment households perceived a higher ability to cope with stress in 11 out of 12 months compared to pregnant women in the control households (Fig 1). Perceived coping in pregnant women in the treatment households was significantly better during the lean season month of May 2020 (*p* = 0.014) and marginally better during the month of June 2020 (*p* = 0.094) (Fig 3). The difference between pregnant women in the treatment and control households was most substantial in September, right before the end of the main harvest of 2020 and the start of the new storage period, when pregnant women in the treatment households perceived themselves to cope significantly better with stress (*p* = 0.002). After the main harvest, pregnant women in the treatment households mostly continued to cope better with stress relative to the control group, although this difference was only significant in December 2020 (*p* = 0.005) (Fig 3). Supplementary S5 Table shows the ITT analysis for all survey months.

## Impact of improved on-farm storage on perceived stress in pregnant women

Pregnant women in the treatment households scored an average of 0.13 points higher than those in the control households on the stress measure over the entire observation period combined, which indicates higher perceived stress (*t*(6082) = 2.314, *p* = 0.021).

When analyzing each survey month separately, pregnant women in the treatment households showed relatively stable perceived stress levels during the lean season, albeit higher compared to those in the control households (Fig 2). This difference was significant for the lean season months of May 2020 (*p* = 0.019), June 2020 (*p* = 0.004), August 2020 (*p* = 0.007) and September 2020 (*p* = 0.037) (Fig 4 and S6 Table). After the end of the main harvest in 2020,

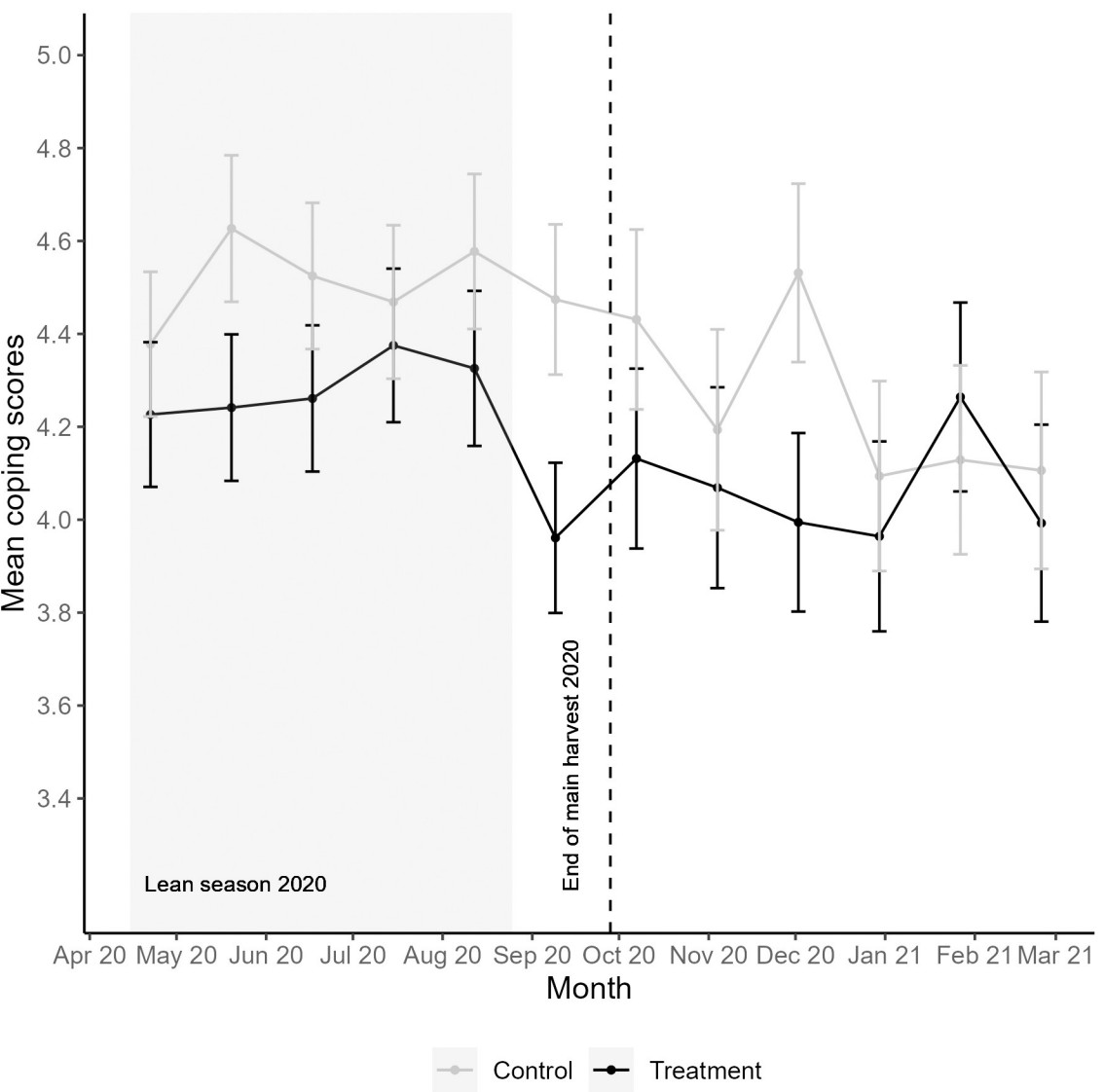

**Fig 1. Mean perceived coping scores and standard error for pregnant women in treatment and control households.** Mean perceived coping is measured by items 2 and 3 of the PSS-4, with a 30-day recall period for all survey months. The weighted mean of all observations per survey month was used to measure the perceived coping factor (see statistical analysis section). Lower mean values indicate higher perceived coping. Vertical bars represent standard errors.

pregnant women in the treatment households more often felt less stressed than those in the control group, but this difference was only marginally significant for the month of January 2021 ($p = 0.086$).

## Discussion

The present study is the first RCT to analyze the causal effect of a food security intervention on perceived stress and perceived coping (component of overall mental health) in pregnant women from smallholder farming households in SSA. Our results show that pregnant women from the treatment households had higher perceived coping scores but experienced more perceived stress compared to pregnant women from the control households.

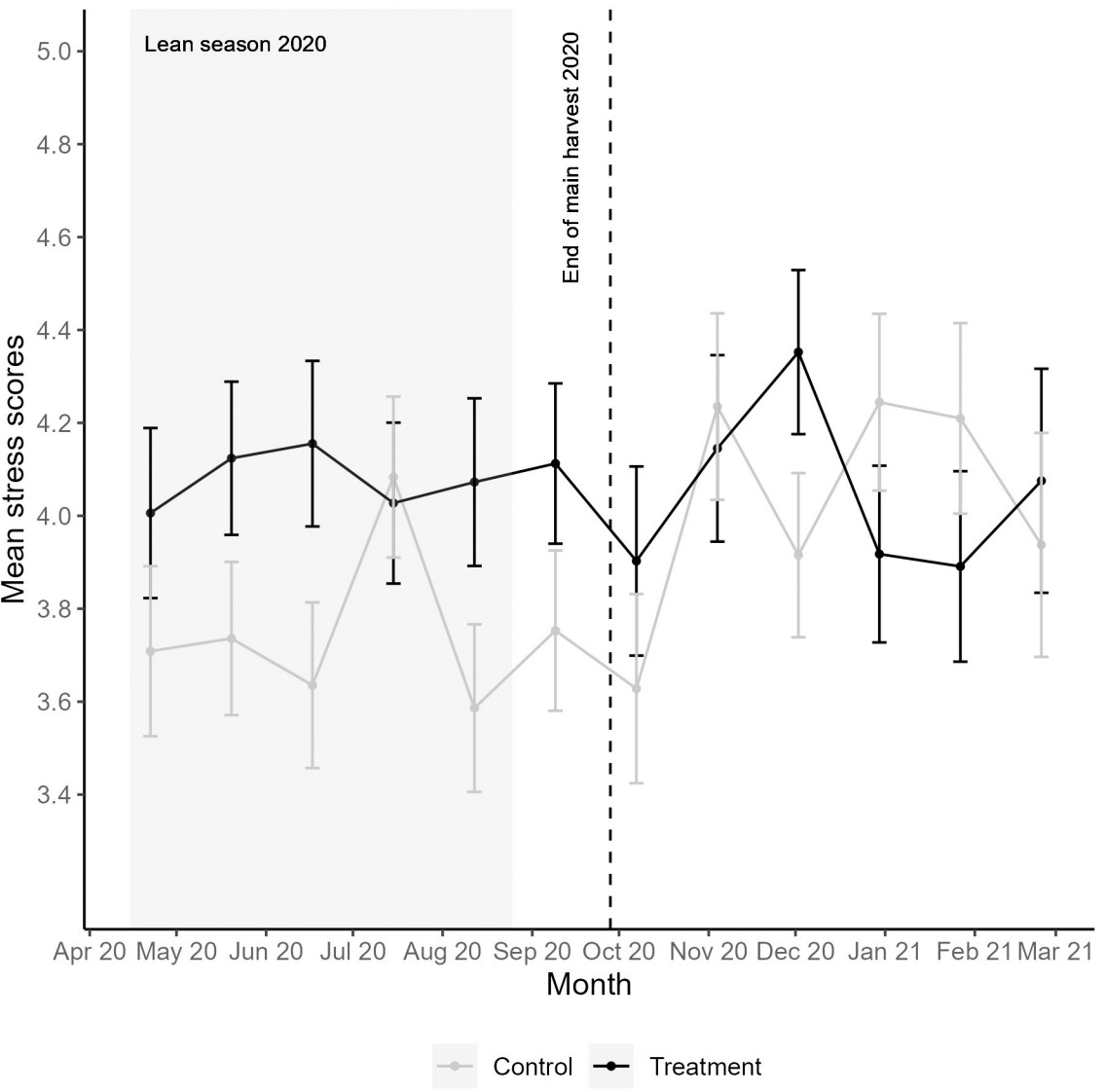

**Fig 2. Mean perceived stress scores and standard errors for pregnant women in treatment and control households.** Mean perceived stress scores are measured by items 1 and 4 of the PSS-4, with a 30-day recall period for all survey months. The weighted mean of all observations per survey month was used to measure perceived stress scores (see statistical analysis section). Higher mean values indicate higher perceived stress. Vertical bars represent standard errors.

The implementation of the hermetic storage bags in the treatment households might have been evaluated as an externally determined environmental stressor that was uncertain, uncontrollable and threatening (primary appraisal), thus triggering a stress response. According to the results of the PSS-4 subscale perceived stress, pregnant women from the treatment households felt less able to control important things in their life and felt that difficulties were piling up so high that they could not overcome them (items 1 and 4) [42]. Previous research has demonstrated that participation in a new economic activity can initially increase psychological distress (measured using the PSS-10 as one factor) [66]. Hence, a lack of experience with this new on-farm storage technology, and uncertainty as to whether and how well these bags would work, may have contributed to the higher perceived stress levels in the treatment households.

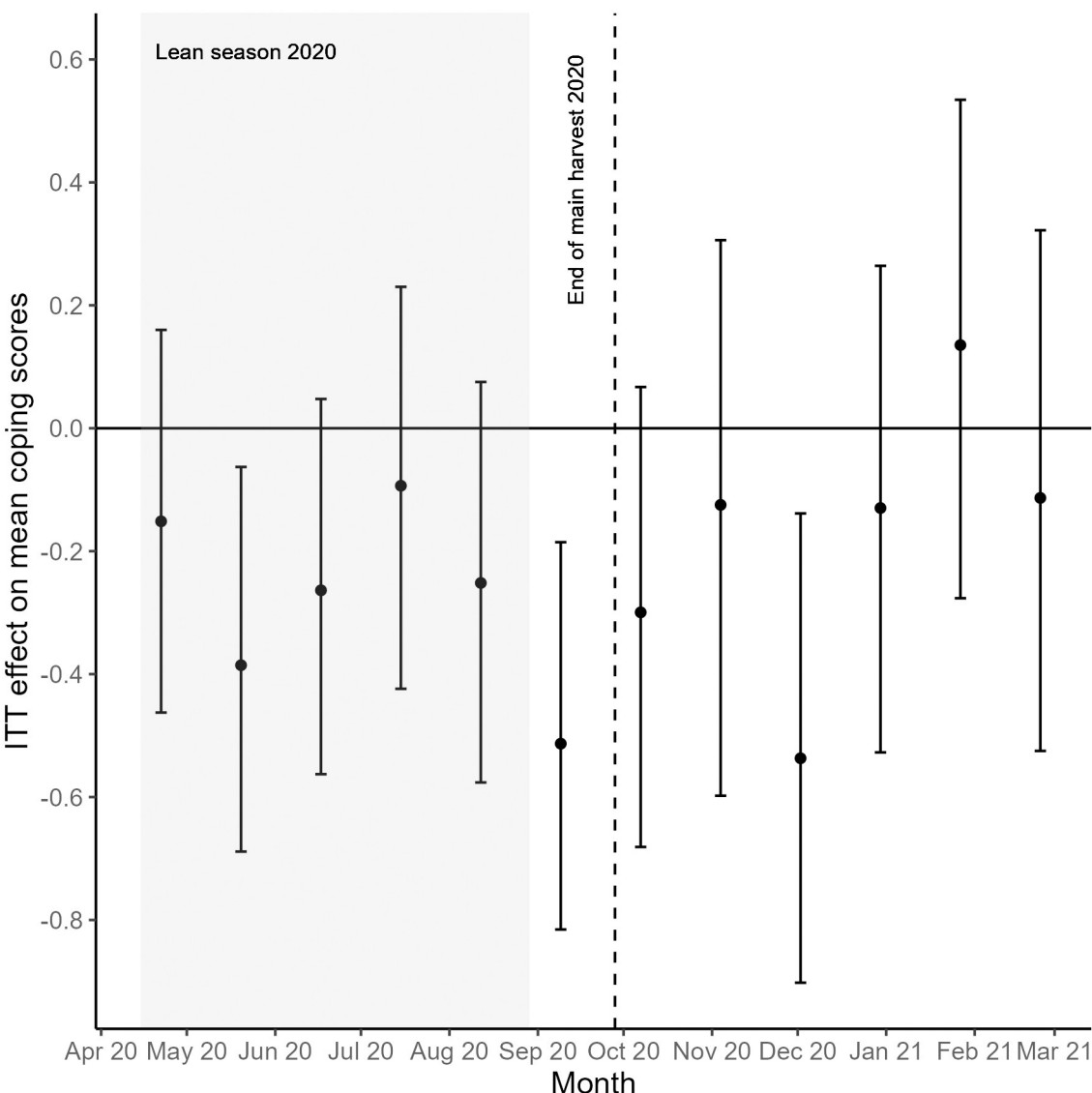

**Fig 3. ITT effects on perceived coping in pregnant women in smallholder farming households.** For each survey month, the points represent the estimated ITT effects on the mean perceived coping scores. Vertical bars show 95% bootstrapped confidence intervals.

As individuals constantly interact with the environment, their cognitive processes (based on past experiences) can change. Therefore, after completion of the first harvest season and having gained the knowledge that these bags reduce food insecurity (adjusted primary appraisal), the significant effects of the treatment on perceived stress in pregnant women diminished and pregnant women in the treatment households therefore tended to be less stressed, albeit not significantly so. Moreover, the uptake of a new agricultural technology that targets food security may take longer to develop a positive impact on perceived stress. Stevenson et al. [67] measured how the implementation of an intervention to improve water quality for Ethiopian households affected psychological distress in Ethiopian women, and found no significant effect on psychological distress measured with the WHO Self-Reporting Questionnaire (SRQ-20). As a possible explanation for this finding, the authors suggested that the time between the

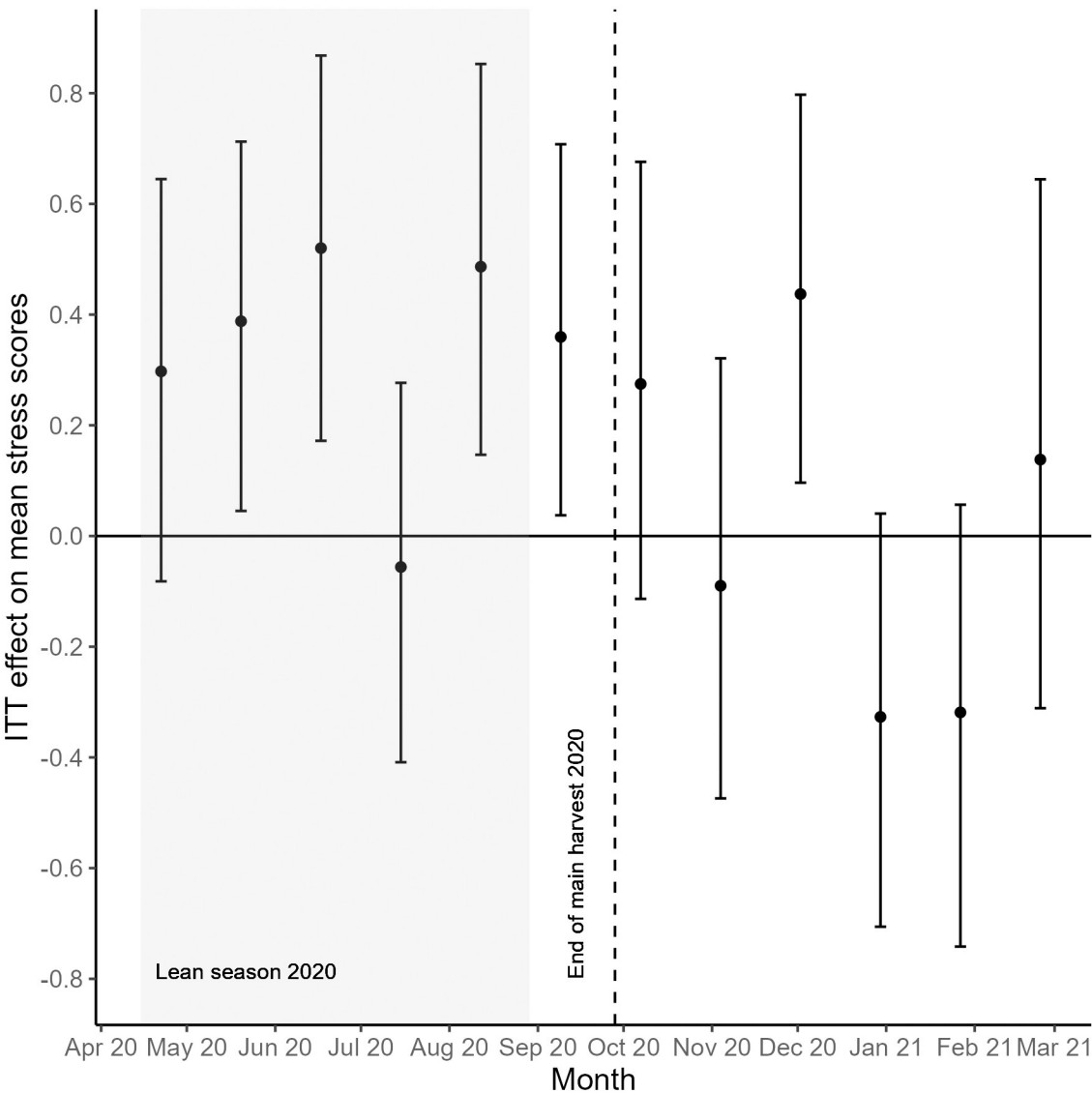

**Fig 4. ITT effects on perceived stress in pregnant women from smallholder farming households.** For each survey round, the points represent the estimated ITT effects on mean perceived stress scores. Vertical bars show 95% bootstrapped confidence intervals.

implementation of the intervention and the stress measurement was too short to have yielded a potential treatment effect. Further research should implement longitudinal panel studies over several years to observe any effects.

Although the implementation of the new technology may pose a perceived threat, the technology itself is a tangible resource (secondary appraisal) that can be used to counteract the effects of PHL and therefore food insecurity. According to the subscale perceived coping of the PSS-4 [42], pregnant women in the treatment households felt more confident about their ability to handle personal problems and felt more that things were going their way (items 2 and 3) compared to pregnant women in the control households. In SSA, women are generally in charge of feeding the household. This forces them to actively overcome food insecurity by adopting different coping strategies such as stretching food resources and skipping meals [68,

69]. Given the higher perceived coping in pregnant women in the treatment households, one might assume that the intervention to reduce PHL empowered these women and gave them a tangible resource, as fewer food coping strategies were required [39]. Empowerment refers to "the process by which those who have been denied the ability to make choices acquire such an ability" [70]. Being deprived of basic needs reduces the ability to make meaningful choices [70]. Accordingly, it seems likely that pregnant women in the control households were deprived of choice over how to successfully feed their families, and this lack of empowerment attributable to food deprivation might have resulted in poorer perceived coping. This finding supports the social causation hypothesis that unfavorable environmental and economic conditions increase the risk of impaired mental health [71, 72].

Mental health in pregnant women is affected by the time of gestation. Recent evidence indicates that the longitudinal course of distress during pregnancy exhibits a U-shaped curve [73], leading to the suggestion that the time of gestation might have impacted the effects of the treatment on perceived stress in our study. However, in contrast to observational studies, our study design reduces the risk of omitted variable bias as the matched-pair design rules out the influence of other potential confounding factors given that potential confounding factors should be similarly distributed in treatment and control households.

It is noteworthy that the timing of data collection coincided with the COVID-19 outbreak and COVID-19 restrictions imposed by the Kenyan government, which began in mid-March 2020 and continued until early July 2020. Focus group discussions with farmers in this RCT indicated that these restrictions had a strong impact on their daily lives, as farmers were unable to access agricultural markets, which in turn affected their food security status (for detailed information, see Huss et al. [39]). Specifically, Huss et al. [39] show that control households' food security declined significantly in the month following COVID-19 restrictions, and much more so than in treatment households. The agricultural intervention mitigated food insecurity among farmers in treatment households [39]. The mitigation of the food shock in treatment households is consistent with our findings that the intervention enables these households to better cope with sudden food shocks. We presume that these effects are randomly distributed among the pregnant women in our study because, to our knowledge, the study region in which our farmer households are located was similarly affected by the COVID-19 crisis. However, it is not possible to systematically disentangle the effect of the intervention and the effect of COVID-19 on our outcome variables.

Some limitations need to be acknowledged when interpreting the findings reported in this study. First, the questionnaire was completed via self-report. However, as there was no personal interaction with the pregnant women, we do not expect a systematic response bias [74]. Second, we cannot guarantee that the pregnant women answered the SMS-based surveys themselves, although to minimize this potential problem we requested in our survey that the PSS-4 should be answered by the pregnant women. Third, it may have been preferable to use a stress questionnaire that better represents the cultural background of the study population, since perceived stress in the PSS-4 might not adequately reflect the local understanding of stress, namely "thinking too much" [75–78].

## Conclusion

This study set out to determine whether improved on-farm storage has positive effects on stress perception and perceived coping abilities among pregnant women from smallholder farming households in Kenya. Results show that pregnant women from treatment households had higher perceived stress levels but also higher perceived coping abilities than pregnant women from control households. The findings suggest that the use of hermetic storage

bags in Kenyan smallholder households provides pregnant women with a sense of empowerment to counteract the effects of PHL and thus food insecurity. Further research could examine the long-term effects of this intervention over several main harvest seasons to determine whether perceived stress levels decrease from one main harvest season to the next as pregnant women gain more experience and confidence in using this new storage technology. In addition, future research should focus on how this storage technology impacts maternal mental health in a broader sense and whether biological mechanisms, such as epigenetics, may underlie this association. Such findings could have important implications for countries in SSA, where women in rural areas are strongly affected by food insecurity and mental health problems [79], and could contribute to the achievement of the second and third goals of the United Nations Agenda for Sustainable Development [80], namely to end food insecurity and ensure well-being and mental health, not just for women but also for their offspring.

## Supporting information

**S1 Table. Translation of the short form of the Perceived Stress Scale (PSS-4).**
(DOCX)

**S2 Table. Cumulative variances of principal component analysis (PCA) for one-factor model and two-factor model.**
(DOCX)

**S3 Table. Minimum and maximum factor loadings for one-factor model and two-factor model.**
(DOCX)

**S4 Table. Sample characteristics of baseline variables between treatment and control group.**
(DOCX)

**S5 Table. Effects of improved on-farm storage on stress coping abilities during pregnancy.**
(DOCX)

**S6 Table. Effects of improved on-farm storage on perceived stress during pregnancy.**
(DOCX)

## Acknowledgments

We thank the International Centre of Insect Physiology and Ecology (*icipe*) for their great collaboration and excellent support in the field.

## Author Contributions

**Conceptualization:** Heike Eichenauer, Thomas Bernauer, Ulrike Ehlert.

**Data curation:** Heike Eichenauer, Matthias Huss, Michael Brander.

**Formal analysis:** Heike Eichenauer, Matthias Huss, Michael Brander.

**Funding acquisition:** Matthias Huss, Michael Brander, Thomas Bernauer, Ulrike Ehlert.

**Methodology:** Heike Eichenauer, Matthias Huss, Michael Brander, Thomas Bernauer, Ulrike Ehlert.

**Project administration:** Heike Eichenauer, Matthias Huss, Michael Brander.

**Supervision:** Thomas Bernauer, Ulrike Ehlert.

**Validation:** Heike Eichenauer.

**Visualization:** Heike Eichenauer.

**Writing – original draft:** Heike Eichenauer.

**Writing – review & editing:** Matthias Huss, Michael Brander, Thomas Bernauer, Ulrike Ehlert.

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
