## [Decision Letter · Decision Letter 0]

30 Mar 2023

PONE-D-23-01447Effects of improved on-farm crop storage on perceived stress and perceived coping in pregnant women – Evidence from a cluster-randomized controlled trial in KenyaPLOS ONE

Dear Dr. Ehlert,

Thank you for submitting your manuscript to PLOS ONE. After careful consideration, we feel that it has merit but does not fully meet PLOS ONE’s publication criteria as it currently stands. Therefore, we invite you to submit a revised version of the manuscript that addresses the points raised during the review process.State the objectives more clearly for readers to follow and understand. Sample selection procedure under the abstract section of the manuscript must be clearly stated.Provide more evidence in the context of the country of study under the introductory section of your manuscript in addition to the sub-Saharan African context. Discussion of findings should be done, putting into consideration the strength and limitations of the study viz-a-viz the period of data collectionWrite in full abbreviations and define clearly at first use e.g RCT, ITT *etc...*Conclusion should emanate from the findings and discussion of the study

We look forward to receiving your revised manuscript.

Kind regards,

Olufunmilayo Olufunmilola Banjo, Ph.D

Academic Editor

PLOS ONE

3. We note that S1 Figure in your submission contain [map/satellite] images which may be copyrighted. All PLOS content is published under the Creative Commons Attribution License (CC BY 4.0), which means that the manuscript, images, and Supporting Information files will be freely available online, and any third party is permitted to access, download, copy, distribute, and use these materials in any way, even commercially, with proper attribution. For these reasons, we cannot publish previously copyrighted maps or satellite images created using proprietary data, such as Google software (Google Maps, Street View, and Earth). For more information, see our copyright guidelines: http://journals.plos.org/plosone/s/licenses-and-copyright.

a. You may seek permission from the original copyright holder of S1 Figure to publish the content specifically under the CC BY 4.0 license. 

Reviewers' comments:

Reviewer's Responses to Questions

**Comments to the Author**

1. Is the manuscript technically sound, and do the data support the conclusions?

Reviewer #1: Yes

Reviewer #2: No

2. Has the statistical analysis been performed appropriately and rigorously? 

Reviewer #1: Yes

Reviewer #2: I Don't Know

3. Have the authors made all data underlying the findings in their manuscript fully available?

Reviewer #1: No

Reviewer #2: No

4. Is the manuscript presented in an intelligible fashion and written in standard English?

Reviewer #1: Yes

Reviewer #2: Yes

5. Review Comments to the Author

Reviewer #1: Effects of improved on-farm crop storage on perceived stress and perceived coping in pregnant women

Abstract

The Abstract is well-written. However, the a very brief and succinct information about the sample can be included in the ‘methodology’ in the Abstract Section.

Introduction

The manuscript is well laid out, and the issues addressed are major development issues pertinent to gender equality, women’s empowerment and health issues. The author provided a well-written and properly review literature showing the linkages of the predictor and outcome variables. However, it will good for the author to provide some evidence in the context of the country covered in this study, Kenya. I acknowledged the fact that the author provided adequate information in the general context of sub-Saharan Africa. However, tailoring this down and providing some available details in the context of Kenya will enrich the information provided. if such details are not available, the authors can state this.

Methods

Methods used were explicitly and clearly explained. The justification for the study design, the choice of the PSS -4 and the statistical analysis conducted were adequately provided. Although I was able to locate a statement on the alternate hypothesis at the last paragraph of the introduction, it will be good to clearly state the null hypothesis, alternate hypothesis etc. in the ‘Methods’ section. Authors can also include in the ‘Methods’ the Research Questions that guided the results obtained from the study.

Discussion

Authors indicated that the observation period for the study was April 2020 to February 2021. This was a period of intense effects of COVID – 19 in many countries of the world…or was Kenya different? This I believed influenced the choice of the data collection method used (SMS based data collection). I am wondering why the authors did not make any mention of possible interference effects of COVID – 19 on the study. The author may also need to include the control measures to isolate such interference. Although in lines 339 – 342, the authors stated ‘in contrast to observational studies, our study design reduces the risk of omitted variable bias as the matched-pair design rules out the influence of other potential confounding factors given that potential confounding factors should be similarly distributed in treatment and control households’. Was this taken care of in the case of possible COVID effects? COVID – 19 was a stressor at the time and could also have had some effects on both the predictor and outcome variables).

Conclusion

The Conclusion section of the manuscript can be revised. Authors can give a very brief and succinct understanding of the study so that readers that are not able to go through the whole study can get a clear concept of the study by reading the conclusion.

Reference

The reference section and references provided look good. Specifications and guidelines provided by PLOS ONE

Table and Figures

Tables and Figures look good especially with the brief information provided after each Table/Figure. Authors can provide a brief explanation also in the S4 Table

Reviewer #2: My main concern of the manuscript is that the authors did not clearly spelt out the objectives of the study as well the conclusion of the study. Only general comments and observations were presented in the conclusion.

6. PLOS authors have the option to publish the peer review history of their article (what does this mean?). If published, this will include your full peer review and any attached files.

Reviewer #1: No

Reviewer #2: No

---

## [Author Response · Author response to Decision Letter 0]

12 May 2023

We would like to thank the editor for raising this point. We ensured that our manuscript meets PLOS ONE’s style requirements, including those for file naming. 

We note that S1 Figure in your submission contain [map/satellite] images which may be copyrighted. We require you to either (1) present written permission from the copyright holder to publish these figures specifically under the CC BY 4.0 license, or (2) remove the figures from your submission. 

We removed Figure S1 from the manuscript and refer to a map of the study region and location of smallholder groups in Huss et al., 2021. Since we removed Figure S1, the other files in the supplementary material were adjusted accordingly. 

We note that you have indicated that data from this study are available upon request. PLOS only allows data to be available upon request if there are legal or ethical restrictions on sharing data publicly. In your revised cover letter, please address the following prompts: If there are ethical or legal restrictions on sharing a de-identified data set, please explain them in detail (e.g., data contain potentially sensitive information, data are owned by a third-party organization, etc.) and who has imposed them (e.g., an ethics committee). Please also provide contact information for a data access committee, ethics committee, or other institutional body to which data requests may be sent.

Thanks for pointing this out and apologies for not making this sufficiently clear. Our data cannot be posted in an internet-based repository that is fully public (openly accessible) because study participants did not give informed consent to such data publication. Our application to the ethics committee (IRB) did not include an explicit request to make the study data publicly available. However, anonymized data will be available upon request to the Ethics committee of ETH Zürich for study replication (Contact details are as follow: Sekretariat Ethikkommission, E-mail: ethics@sl.ethz.ch) or upon request to the corresponding author (Prof. rer. nat. Ulrike Ehlert, E-mail: u.ehlert@psychologie.uzh.ch). The revised information about data sharing is also included in the revised cover letter. 

Reviewer #1 

We are grateful for the positive assessment of our study and thank the reviewer for her/his highly valuable feedback. In the following, we respond to the points raised by the reviewer. 

1. The Abstract is well-written. However, a very brief and succinct information about the sample can be included in the ‘methodology’ in the Abstract Section.

We have added information about the sample selection in the Methods section of the Abstract. 

In a randomized controlled trial (RCT), short message service (SMS)-based mobile phone surveys were conducted to assess the causal effect of a food security intervention (improved on-farm storage of maize) on perceived stress and coping in pregnant women from smallholder households. Pregnant women were identified through these monthly surveys by asking whether someone in their household was currently pregnant.

2. Introduction: The manuscript is well laid out, and the issues addressed are major development issues pertinent to gender equality, women’s empowerment and health issues. The author provided a well-written and properly review literature showing the linkages of the predictor and outcome variables. However, it will good for the author to provide some evidence in the context of the country covered in this study, Kenya. I acknowledged the fact that the author provided adequate information in the general context of sub-Saharan Africa. However, tailoring this down and providing some available details in the context of Kenya will enrich the information provided. if such details are not available, the authors can state this.

Thanks for raising this important point and for giving us the opportunity to tailor the review literature more closely to the context of the country covered in this study. We have added this information in the second and third paragraph of the introduction. 

… With regard to food insecurity, numerous studies have reported a positive association between increased food insecurity and impaired mental health in African populations [14]. For instance, pregnant Ethiopian women living in food-insecure households were four times more likely to experience psychological distress than their food-secure counterparts [15]. Psychological distress is an important component of overall mental health and a risk factor for depression [16–18]. Researchers have demonstrated that food-insecure pregnant women in Africa are up to five times more likely to suffer from depression [19,20]. In the context of the country examined in this study, Kenya, there is evidence of a positive association between food insecurity and depressive symptoms in perinatal women [21,22]. Specifically, the odds of suffering from depressive symptoms increased almost six fold when food insecurity was high [21]. In addition, perceived stress was higher among Kenyan postpartum women who experienced moderate food insecurity than among Kenyan postpartum women who experienced mild food insecurity [23]. 

… Furthermore, heavy rainfalls can prevent smallholder farmers from adequately drying their crops before storage, which can lead to increased moisture and thus mold growth during the storage period, rendering the crop inedible and therefore lost [29–31]. Extreme weather shocks, such as the aforementioned drought and rainfall, negatively affect crop production in Kenya [32] and are a major concern for smallholder farmers in rural Kenya [33]. Post-harvest losses (PHL) refer to the amount of crop lost due to factors such as crop handling and crop storage [31], with the latter accounting for the majority of PHL [34,35]. Indeed, a meta-analysis revealed that on average, 25.6% of harvested maize in SSA is lost due to insufficient crop storage [35]. A recent article states that the PHL of maize by storage pests amount to about 36% overall, with Western Kenya being the most affected [36]. … 

3. Methods used were explicitly and clearly explained. The justification for the study design, the choice of the PSS -4 and the statistical analysis conducted were adequately provided. Although I was able to locate a statement on the alternate hypothesis at the last paragraph of the introduction, it will be good to clearly state the null hypothesis, alternate hypothesis etc. in the ‘Methods’ section. Authors can also include in the ‘Methods’ the Research Questions that guided the results obtained from the study.

We have added the research question as well as alternative and null hypothesis in the Methods section under the sub-section Research aim and statistical analysis. 

Research aim and statistical analysis 

The purpose of this study was to determine whether an agricultural intervention consisting of an improved on-farm storage technology with hermetic storage bags affects the perceived stress levels and coping abilities of pregnant women in Kenyan smallholder households. As an alternative hypothesis, we also explore whether pregnant women in the treatment households have lower perceived stress and higher perceived coping abilities compared with pregnant women in the control households. Our null hypothesis is that there are no differences between perceived stress and coping by pregnant women in treatment and control households. 

ITT analyses was used to analyze treatment effects (allocation and use of hermetic storage bags in the household) on perceived stress and perceived coping in pregnant women from smallholder farming households. … 

4. Discussion: Authors indicated that the observation period for the study was April 2020 to February 2021. This was a period of intense effects of COVID – 19 in many countries of the world…or was Kenya different? This I believed influenced the choice of the data collection method used (SMS based data collection). I am wondering why the authors did not make any mention of possible interference effects of COVID – 19 on the study. The author may also need to include the control measures to isolate such interference. Although in lines 339 – 342, the authors stated ‘in contrast to observational studies, our study design reduces the risk of omitted variable bias as the matched-pair design rules out the influence of other potential confounding factors given that potential confounding factors should be similarly distributed in treatment and control households’. Was this taken care of in the case of possible COVID effects? COVID – 19 was a stressor at the time and could also have had some effects on both the predictor and outcome variables).

Thanks a lot for raising this point. The COVID-19 outbreak did not affect the choice of data collection method used. SMS-based data collection was always our first choice because of its cost-effective and advantageous real-time high-frequency nature. In addition, our study participants were geographically dispersed, which would have made face-to-face interviews extremely costly and logistically impossible over several months. It is true that COVID-19 was a stressor at the time of data collection and therefore may have had some impact on the predictor and outcome variables. We have addressed this concern in paragraph 5 of the discussion.

It is noteworthy that the timing of data collection coincided with the COVID-19 outbreak and COVID-19 restrictions imposed by the Kenyan government, which began in mid-March 2020 and continued until early July 2020. Focus group discussions with farmers in this RCT indicated that these restrictions had a strong impact on their daily lives, as farmers were unable to access agricultural markets, which in turn affected their food security status (for detailed information, see Huss et al. [39]). Specifically, Huss et al. [39] show that control households’ food security declined significantly in the month following COVID-19 restrictions, and much more so than in treatment households. The agricultural intervention mitigated food insecurity among farmers in treatment households [39]. The mitigation of the food shock in treatment households is consistent with our findings that the intervention enables these households to better cope with sudden food shocks. We presume that these effects are randomly distributed among the pregnant women in our study because, to our knowledge, the study region in which our farmer households are located was similarly affected by the COVID-19 crisis. However, it is not possible to systematically disentangle the effect of the intervention and the effect of COVID-19 on our outcome variables.

5. The Conclusion section of the manuscript can be revised. Authors can give a very brief and succinct understanding of the study so that readers that are not able to go through the whole study can get a clear concept of the study by reading the conclusion.

We agree and have revised the concluding section of the manuscript. 

This study set out to determine whether improved on-farm crop storage has positive effects on stress perception and perceived coping abilities among pregnant women from smallholder households in Kenya. Results show that pregnant women from treatment households had higher perceived stress levels but also higher perceived coping abilities than pregnant women from control households. These findings suggest that the use of hermetic storage bags in Kenyan smallholder households provides pregnant women with a sense of empowerment to counteract the effects of PHL and thus food insecurity. Further research could examine the long-term effects of this intervention over several main harvest seasons to determine whether perceived stress levels decrease from one main harvest season to the next as pregnant women gain more experience and confidence in using this new storage technology. In addition, future research should focus on how this storage technology impacts maternal mental health in a broader sense and whether biological mechanisms, such as epigenetics, may underlie this association. Such findings could have important implications for countries in SSA, where women in rural areas are strongly affected by food insecurity and mental health problems [79], and could contribute to the achievement of the second and third goals of the United Nations Agenda for Sustainable Development [80], namely to end food insecurity and ensure well-being and mental health, not just for women but also for their offspring. 

6. The reference section and references provided look good. Specifications and guidelines provided by PLOS ONE

We would like to thank the reviewer for this feedback.

7. Tables and Figures look good especially with the brief information provided after each Table/Figure. Authors can provide a brief explanation also in the S4 Table

We have added a brief explanation in the S3 Table (S3 due to the removal of Figure S1 - all supplementary material has been adjusted accordingly). 

Reviewer #2

We are grateful for this assessment of the study and thank the reviewer for her/his valuable feedback.

My main concern of the manuscript is that the authors did not clearly spelt out the objectives of the study as well the conclusion of the study. Only general comments and observations were presented in the conclusion.

Thanks for raising this point and for giving us the opportunity to spell out the objectives of the study more clearly and revise the conclusion to make it more precise. We have added information regarding the objectives in the abstract and in the last paragraph of the introduction and in the added sub-section “Research aim” in the Methods section. In addition, we have revised the conclusion in the abstract and in the manuscript. 

Background section of the abstract 

… However, it remains to be determined whether this agricultural intervention has an additional effect on pregnant women’s perceived stress levels and coping abilities. This study examines whether pregnant women from treatment households experience lower perceived stress levels and higher perceived coping abilities compared to pregnant women from control housholds.

Last paragraph of the introduction

In summary, on the correlation level, numerous studies have shown that food insecurity is associated with poorer mental health outcomes. We take this research forward by determine whether an agricultural intervention to improve food security (allocation and use of hermetic storage bags in households) can have beneficial mental health outcomes for pregnant women from smallholder households. We hypothesized that pregnant women in the intervention households (treatment) would experience lower perceived stress and higher perceived coping as compared to pregnant women in the control households. Following the randomly allocated intervention, we collected perceived stress and perceived coping data over several months using monthly short message service (SMS)-based surveys. The treatment effects were analyzed via Intention-to-treat (ITT) analyses. 

Research aim and statistical analysis 

The purpose of this study was to determine whether an agricultural intervention consisting of an improved on-farm storage technology with hermetic storage bags affects the perceived stress levels and coping abilities of pregnant women in Kenyan smallholder households. As an alternative hypothesis, we also explore whether pregnant women in the treatment households have lower perceived stress and higher perceived coping abilities compared with pregnant women in the control households. Our null hypothesis is that there are no differences between perceived stress and coping by pregnant women in treatment and control households. 

ITT analyses was used to analyze treatment effects (allocation and use of hermetic storage bags in the household) on perceived stress and perceived coping in pregnant women from smallholder farming households. … 

Revised conclusion of the abstract 

Our findings indicate that pregnant women from treatment households had higher perceived coping abilities but experienced more perceived stress. More research is needed on how this intervention impacts maternal mental health and whether biological mechanisms, such as epigenetics, may underlie this association. 

This study set out to determine whether improved on-farm crop storage has positive effects on stress perception and perceived coping abilities among pregnant women from smallholder households in Kenya. Results show that pregnant women from treatment households had higher perceived stress levels but also higher perceived coping abilities than pregnant women from control households. These findings suggest that the use of hermetic storage bags in Kenyan smallholder households provides pregnant women with a sense of empowerment to counteract the effects of PHL and thus food insecurity. Further research could examine the long-term effects of this intervention over several main harvest seasons to determine whether perceived stress levels decrease from one main harvest season to the next as pregnant women gain more experience and confidence in using this new storage technology. In addition, future research should focus on how this storage technology impacts maternal mental health in a broader sense and whether biological mechanisms, such as epigenetics, may underlie this association. Such findings could have important implications for countries in SSA, where women in rural areas are strongly affected by food insecurity and mental health problems [79], and could contribute to the achievement of the second and third goals of the United Nations Agenda for Sustainable Development [80], namely to end food insecurity and ensure well-being and mental health, not just for women but also for their offspring. 

We would like to thank all reviewers again for their highly valuable feedback.

---

## [Decision Letter · Decision Letter 1]

28 Jun 2023

Effects of improved on-farm crop storage on perceived stress and perceived coping in pregnant women – Evidence from a cluster-randomized controlled trial in Kenya

PONE-D-23-01447R1

Dear Dr. Ulrike Ehlert,

We’re pleased to inform you that your manuscript has been judged scientifically suitable for publication and will be formally accepted for publication once it meets all outstanding technical requirements.

Kind regards,

Olufunmilayo Olufunmilola Banjo, Ph.D

Academic Editor

PLOS ONE

Additional Editor Comments (optional):

Reviewers' comments:

Reviewer's Responses to Questions

**Comments to the Author**

1. If the authors have adequately addressed your comments raised in a previous round of review and you feel that this manuscript is now acceptable for publication, you may indicate that here to bypass the “Comments to the Author” section, enter your conflict of interest statement in the “Confidential to Editor” section, and submit your "Accept" recommendation.

Reviewer #1: All comments have been addressed

2. Is the manuscript technically sound, and do the data support the conclusions?

Reviewer #1: Yes

3. Has the statistical analysis been performed appropriately and rigorously? 

Reviewer #1: Yes

4. Have the authors made all data underlying the findings in their manuscript fully available?

Reviewer #1: No

5. Is the manuscript presented in an intelligible fashion and written in standard English?

Reviewer #1: Yes

6. Review Comments to the Author

Reviewer #1: (No Response)

7. PLOS authors have the option to publish the peer review history of their article (what does this mean?). If published, this will include your full peer review and any attached files.

Reviewer #1: No

---

## [Editor Report · Acceptance letter]

5 Jul 2023

PONE-D-23-01447R1 

Effects of improved on-farm crop storage on perceived stress and perceived coping in pregnant women – Evidence from a cluster-randomized controlled trial in Kenya 

Dear Dr. Ehlert:

I'm pleased to inform you that your manuscript has been deemed suitable for publication in PLOS ONE. Congratulations! Your manuscript is now with our production department. 

Kind regards, 

on behalf of

Dr. Olufunmilayo Olufunmilola Banjo 

Academic Editor

PLOS ONE